# Effects of the Acetyltransferase p300 on Tumour Regulation from the Novel Perspective of Posttranslational Protein Modification

**DOI:** 10.3390/biom13030417

**Published:** 2023-02-22

**Authors:** Qingmei Zeng, Kun Wang, Yongxiang Zhao, Qingzhi Ma, Zhinan Chen, Wan Huang

**Affiliations:** 1National Center for International Research of Bio-Targeting Theranostics, Guangxi Key Laboratory of Bio-Targeting Theranostics, Collaborative Innovation Center for Targeting Tumor Diagnosis and Therapy, Guangxi Talent Highland of Bio-Targeting Theranostics, Guangxi Medical University, Nanning 530021, China; 2National Translational Science Center for Molecular Medicine, Department of Cell Biology, Fourth Military Medical University, Xi’an 710032, China

**Keywords:** p300, acetyltransferase, tumours, modification, drug resistance, inhibitor

## Abstract

p300 acts as a transcription coactivator and an acetyltransferase that plays an important role in tumourigenesis and progression. In previous studies, it has been confirmed that p300 is an important regulator in regulating the evolution of malignant tumours and it also has extensive functions. From the perspective of non-posttranslational modification, it has been proven that p300 can participate in regulating many pathophysiological processes, such as activating oncogene transcription, promoting tumour cell growth, inducing apoptosis, regulating immune function and affecting embryo development. In recent years, p300 has been found to act as an acetyltransferase that catalyses a variety of protein modification types, such as acetylation, propanylation, butyylation, 2-hydroxyisobutyration, and lactylation. Under the catalysis of this acetyltransferase, it plays its crucial tumourigenic driving role in many malignant tumours. Therefore, the function of p300 acetyltransferase has gradually become a research hotspot. From a posttranslational modification perspective, p300 is involved in the activation of multiple transcription factors and additional processes that promote malignant biological behaviours, such as tumour cell proliferation, migration, and invasion, as well as tumour cell apoptosis, drug resistance, and metabolism. Inhibitors of p300 have been developed and are expected to become novel anticancer drugs for several malignancies. We review the characteristics of the p300 protein and its functional role in tumour from the posttranslational modification perspective, as well as the current status of p300-related inhibitor research, with a view to gaining a comprehensive understanding of p300.

## 1. Introduction

Since it was discovered that p300 plays a role as an auxiliary transcription activator, many researchers have begun to focus on p300, specifically its role in tumours. p300 can function as a cancer-promoting factor to regulate various biological functions and the malignant progression of various cancers, including lung cancer, stomach cancer, colorectal carcinoma, oesophageal squamous cell carcinoma (ESCC), pancreatic cancer and prostate cancer [1,2,3,4,5,6,7]. In addition, p300 can play an antitumour role in epidermal tumours, osteosarcoma, myelodysplastic syndrome (MDS)-associated leukaemia and human papilloma virus (HPV)-positive head and neck squamous cell carcinoma (HNSCC) [8,9,10,11], and tumours with high EP300 mutation rates have enhanced antitumour immunity [12]. p300 also participates in the regulation of some nontumour diseases and processes, including viral infections [13], chronic constriction injury (CCI) [14], and some cardiovascular diseases and processes (hypertension [15], heart development and heart ageing [16]).

p300 has extensive functions, and it is a common hub of many important biological pathways [17]. For example, p300 can participate in the regulation of many signalling pathways related to p53 [18], the oestrogen regulatory pathway [19], the ATR-CHK1 axis [20] and the cAMP pathway [21]. p300 can also bind to other proteins, such as HIF-α [22], β-catenin [23], p53 [24], TR3 [25], Myb [26] and Smad2/Smad3 [27], and play integral roles in complex physiological and pathological processes. Moreover, p300 expression is modulated by other molecules, such as RAR and ATRA [28]. There is a new global focus on the acetyltransferase activity of p300, and it has been recognised that p300 plays a crucial role in the malignant development of tumours by mediating the development of different types of modifications of histones and nonhistone proteins; these functions have presented new opportunities to expand the knowledge on p300 functions.

## 2. The Structure of the p300 Protein

p300 was first discovered in 1986 and is a protein with a molecular weight of 300 kDa. p300 is also the nuclear binding target of the adenovirus E1A cancer protein. It has high structural homology with CBP, with an approximately 70% homologous sequence. CBP is an important protein that can bind to cAMP reaction element binding protein (CREB). p300 and CBP have similar but different functions; for example, p300 is mainly involved in transcription and mediates posttranslational modification, while CBP mainly plays a role in cell transformation. The p300 structural domain mainly includes the N-terminal nuclear receptor interaction domain (NRID), transcription adapter zinc finger 1 (TAZ1) and the kinase-induced CREB interaction (KIX) domain [29,30]. The HAT domain is very important for the formation of the transcription complex. Furthermore, it also contains a bromine domain (BD): this domain is the key to the auxiliary transcription function and is involved in the interaction of acetylated modified proteins and the regulation of different physiological and pathological processes; it is also involved in the interaction of the RING domain with the plant homologous domain (PHD), which leads to interactions with the ZZ-type zinc finger domain, TAZ2 domain and IBiD domain [31,32,33], as shown in Figure 1.

## 3. p300 and Posttranslational Protein Modification

Posttranslational modification of proteins, a form of posttranslational protein processing, involves covalent binding of a small molecular group or a small protein to a specific position in the amino acid side chain of a protein or enzymatic digestion of a protein that causes a break in the covalent bonds of its main chain [34]. The types of posttranslational protein modifications include oxidative modification [35], phosphorylation [36], ubiquitination [37], methylation [38], formylation [39], acetylation [40], propionylation [41], malonylation [42], butyrylation [43], glutarylation [44], succinylation [45], crotonylation [46], and lactylation [47]. Posttranslational protein modifications are associated with different physiological states and the onset of multiple diseases. The functions of posttranslational modifications are diverse and can include the regulation of transcriptional processes, protein–protein interactions, protein localisation and malignant tumour progression.

In particular, p300 can participate in the regulation of posttranslational protein modification by acting as an acetyltransferase and catalyses different types of specific protein modifications, including acetylation, propionylation, butyrylation, n-butyrylation, isobutyrylation, β-hydroxybutyrylation, 2-hydroxyisobutyrylation, crotonylation, isonicotinylation, and lactylation [43,47,48,49,50,51,52,53,54,55,56] (Table 1). p300 itself can also undergo autoacetylation, propionylation, and butyrylation [57,58], and it plays an indispensable role in a variety of physiological and pathological processes, one of which is regulating the malignant progression of tumours. The following figure shows the experimental schematic diagram of mass spectrometry detecting a protein’s posttranslation modification (Figure 2).

## 4. p300 and Tumours

### 4.1. p300 Can Regulate Transcription

Transcription is the process of transforming genetic information from DNA to RNA. As the first step of protein biosynthesis, transcription is the synthesis step of mRNA and non-coding RNA (tRNA, rRNA, etc.). p300 can regulate transcription, acting as a coactivator of transcription through two mechanisms. First, p300 regulates transcription factor activity, binds enhancers and promotes gene expression by exerting its own transcriptional functions or by cooperating with other molecules. Secondly, p300 regulates transcription factors by promoting modifications that affect the transcription process. The second mechanism has been the focus of recent studies.

p300 can function as an acetyltransferase to modify certain proteins and then participate in the transcription process. For instance, in gastric cancer and melanoma, p300 affects the proliferation, cell cycle and ageing of tumour cells to some extent by acetylating histones to promote transcription of the corresponding genes [59,60], and a similar finding was found in diffuse large B-cell lymphoma (DLBCL) [61]. p300 also regulates transcriptional processes in prostate cancer; on the one hand, p300 recognise phosphorylated AR, which promotes subsequent AR acetylation at K609, thereby facilitating transcription [62]. On the other hand, p300 can promote the malignant progression of prostate cancer by increasing the levels of H3K18ac, H3K27ac and H4ac in the transcription start site (TSS) and antioxidant response element (ARE) regions, which in turn affects the transcriptional level [63]. In breast cancer, the p300 inhibitor A485 can decrease the transcript levels of highly expressed genes by reducing the H3K27ac level in specific genes, such as ER [64]. It has also been demonstrated that DOT1L interacts with the p300/c-Myc complex to enhance EMT-induced stemness properties by recognising promoters such as those of ZEB1 and ZEB2; this promotes DNA methylation and histone acetylation, enhances epithelial−mesenchymal transition (EMT) regulators, and accelerates the malignant progression of breast cancer [65]. Regarding esophageal adenocarcinoma, p300 acetylates K188 and K189 of Maml1, recruiting NACK into the Notch1 ternary complex, which in turn leads to transcription initiation [66]. The interaction of p300 with MRTF-A was verified in MCF-7 cells, and p300 was found to be recruited to the promoter of MRTF-A target genes to acetylate histones and thereby reconfigure chromatin structure [67]. In addition, p300 can play a role in tumour immune regulation. p300 enhances myeloid-derived suppressor cell (MDSC) immunosuppressive function by regulating C/EBPβ acetylation and Arg-1 transcription [68].

Studies on the effect of p300 inhibitors on transcriptional regulation have found that treatment of multiple myeloma cells with A485 preferentially enables the deacetylation of enhancer progenitor-associated histones at the transcriptional level [69]. Similar findings were observed in human colorectal carcinomas [6]. The p300 inhibitor A485 inhibits acetylation of H3 at the CD274 (encoding PD-L1) promoter site and blocks CD274 transcription, providing a mechanistic basis for the combination of A485 and anti-PD-L1 antibodies in the treatment of prostate cancer [70].Treatment of hepatocellular carcinoma (HCC) with the p300 inhibitor B029-2 inhibits the transcription of six metabolic genes, PSPH, PSAT1, ALDH18A1, ATIC, TALDO1 and DTYMK by blocking the binding of H3K18ac and H3K27Ac to the relevant promoters [71]. In macrophages and induced pluripotent stem cells (iPSCs), p300 has been shown to regulate histone lactylation, and p300 is known to catalyse YTHDF2 promoter lactylation in ocular melanoma cells. HDAC1-3 and SIRT1-3 were shown to be demulsification in other studies [72,73] (Table 2). Figure 3 shows a schematic representation of transcription-related pathways regulated by p300.

### 4.2. p300 Regulates Tumour Cell Proliferation, Migration, and Invasion

p300, a tumour-promoting protein, plays an essential role in malignant progression in most solid tumours, and its role has been confirmed in prostate cancer, colorectal cancer, liver cancer, and other diseases. Many years of research have shown that p300 can be involved in biological processes that regulate tumour cell proliferation, migration and invasion, Tumour cell proliferation, migration, and invasion are three important features of the biological behaviour of malignant tumour development. Of these, tumour cell proliferation is the basis of tumour cell growth, development, inheritance and reproduction, and cells all proliferate by division. Tumour cell migration, also known as tumour cell crawling, is a common phenomenon in the pathological process of tumour metastasis. Cellular invasion is the process of expansion and proliferation of malignant tumour cells from their origin to surrounding normal tissue along the interstitial space. Indicating that tumour cells breach the basement membrane. and a clear understanding of the p300 mechanism of action provides a solid basis for further clarification of the role of p300. Furthermore, the focus of research on p300 over the years has shifted, and most studies now focus on the role of p300 as an acetylase.

p300 possesses acetyltransferase activity and plays a procancer role in both solid and haematological cancers. In many solid tumours, such as breast cancer [89], hepatocellular carcinoma [90], oesophageal cancer [91], and cutaneous squamous cell carcinomas [92], increased p300 expression is associated with an aggressive phenotype and a low survival rate; furthermore, histochemistry analysis of a variety of primary tumour tissues has revealed that lower overall levels of H3K18ac are associated with lower survival and a higher risk of recurrence in prostate cancer, renal cancer, lung cancer, pancreatic cancer and breast cancer [93,94,95,96,97]. Wang et al. showed that p300 can acetylate PHF5A, promote cell proliferation, and affect the prognosis of colorectal cancer [98]. In studies of colorectal cancer development, p300 was found to act not only through overexpression of the Wnt signalling pathway but also through interaction with and acetylation of MTA2 to promote migration and invasion [74,99]. HAN et al. suggested that cervical cancer is highly malignant in part because p300 promotes the proliferation, migration, and invasion of HeLa cells by mediating lysine crotonylation and increasing HNRNPA1 expression [79]. p300 can promote the invasion and migration of human nasopharyngeal carcinoma cells (CNE-2 cells), possibly by acetylating Smad2 and Smad3 through the TGF-β signalling pathway and inducing EMT [76]. AR is acetylated, and its stability is regulated by p300, which promotes prostate cancer cell proliferation [77]. In MCF-7 cell experiments, overexpression of p300 and MRTF-A was found to catalyse MRTF-A acetylation and to increase the transcript levels of migration-related genes such as MYL9, MYH9 and CYR61, which in turn increased breast cancer cell viability and promoted cell migration [67]. However, p300 can exert a tumour suppressor effect; for example, in regulating osteosarcoma development, p300 can acetylate JHDMIA K409 and can inhibit the proliferation and invasion of HOS osteosarcoma cells [11].

Regarding haematological tumours, the p300 inhibitor C646 was found to cause apoptosis and slow the proliferation of AML1-ETO-positive acute myeloid leukaemia (AML) cells. Treatment with C464 delayed the growth and induced apoptosis of AML1-ETO-positive AML cell lines and primary parental cells, and the study found that acetylation of AML1-ETO by p300 may be responsible [100,101]. In T-cell acute lymphoblastic leukaemia, p300 and HDAC1 were found to acetylate and deacetylate the K1692 and K1731 positions of Notch3, which plays a key role in T-cell proliferation and the overall malignant progression of T-cell leukaemia [81]. In human leukaemia cell lines, it was also demonstrated that the CREBBP/EP300 bromodomain is essential for regulating the GATA1/MYC regulatory axis during proliferation, and administration of CREBBP/EP300 bromodomain inhibitors was very effective in reducing H3K27ac levels and thus hindering cancer cell proliferation [102].

In successive studies exploring the therapeutic feasibility of p300 inhibitors, it was found that A485, a p300-specific inhibitor, significantly inhibited the growth of ER+ breast cancer cells, which may be related to the down-regulation of enhancer H3K27ac, which inhibits the expression of ER target gene [64]. The antitumour activity of A485 in growth hormone pituitary adenoma was also associated with reduced levels of H3K18ac and H3K27ac [103]. Treatment of colon cancer cells with the isothiazolone-based agent PCAF and the p300 histone acetyltransferase (HAT) co-inhibitors CCT077791 and CCT077792 inhibited histone acetylation and thus prevented colon cancer cell proliferation [104,105]. Treatment of hepatocellular carcinoma cells with the highly potent p300 inhibitor B029-2 significantly reduced the levels of H3K18ac and H3K27ac and significantly reduced the proliferation and metastatic capacity of hepatocellular carcinoma cells [71]. The reason why the combination of curcumin and anti-PD-1 therapy is more effective than anti-PD-1 therapy alone is that curcumin can reduce p300-induced histone acetylation in the promoter region of TGF-β1 because it inhibits p300 expression, thereby activating immune cell function and reducing immune escape [106]. Triple-negative breast cancer cells are sensitive to the p300 inhibitor L002, and treatment with L002 can significantly inhibit the growth of cancer cells. In animal experiments, it was also confirmed that L002 could significantly inhibit tumour growth in vivo, and histochemistry experiments showed that the level of H4ac was significantly decreased in the group that received L002. It was further speculated that L002 might function by decreasing the level of histone acetylation [107]. In summary, p300 is a potential target for the treatment of tumours because it can regulate biological processes such as tumour cell growth, migration and invasion (Table 2), and these processes are presented in Table 2.

### 4.3. p300 Regulates Tumour Cell Apoptosis

Tumour cell apoptosis is an ordered or procedural manner of cell death, and it is an active death process of tumour cells under the control of specific genes. Apoptotic cells will ultimately be processed by phagocytes. In the case of cancer cells, the interruption of the apoptosis process signifies the development and spread of cancer. Apoptosis significantly affects the fate of tumour cells, and in general, p300 directly and indirectly regulates tumour cell apoptosis and affects tumour progression. p300 affects the following regulatory signalling molecules during the complex pathogenesis of the development of different tumours: c-MYC [108], c-Met, cyclin D1, Bcl2 [2], TRAIL [109], RAR and ATRA [28], Wnt/β-catenin [74], API5 [110], p53 [111], etc. p300 acts as an acetyltransferase, catalysing the acetylation of histones and nonhistone proteins, and plays an integral role in regulating tumour cell apoptosis. Ono et al. performed pancreatic cancer cell-related experiments and found that the expression of apoptosis-related proteins such as cleaved caspases 3, 8, and 9 and PARP was increased after p300 interference, and acetylation of H3K27 was also inhibited by C646 treatment, suggesting that p300 may regulate apoptosis of pancreatic cancer cells via its HAT activity [83]. Liu et al. found that p300 is involved in the acetylation of H3 and H4 on the RASSF2A promoter and regulates RASSF2A expression, which in turn induces the apoptosis of gastric cancer cells [59]. In studies by Fu and others, C646 treatment led to the apoptosis of AML1-ETO-positive AML cell lines and primary parental cells, while normal haematopoietic stem cells were not affected [100,112]. Because p300 can mediate histone acetylation, it is speculated that targeting p300 to regulate histone acetylation may be an important new therapeutic option for AML treatment [101]. Interestingly, p300 can also catalyse the modification of some nonhistone proteins; for example, in the presence of hypoxia or DNA damage, p300 can acetylate p53, which in turn regulates apoptosis [113,114] (Table 2). Figure 4 shows a schematic representation of the pathways involved in the regulation of apoptosis by p300.

### 4.4. p300 Regulates the Formation of Tumour Drug Resistance

Drug resistance, leading to a decrease or lack of efficacy of the drug towards the pathogen, generally refers to the decline or even disappearance of the pathogen’s susceptibility to the drug following repeated drug contact. The emergence of drug resistance is a significant reason for the ineffectiveness of tumour treatment, and it is a problem that needs to be understood and solved. As a regulator closely related to the degree of tumour malignancy, p300 is also involved in the development of drug resistance. Lapatinib is a chemotherapeutic drug for breast cancer, and p300 mediates FOXO3 acetylation and enhances sensitivity to lapatinib [84]. It has also been demonstrated in relevant experiments in cisplatin-resistant bladder cancer cells that cisplatin resistance may be acquired by p300 catalysing foxo3a acetylation[115]. Ono et al. found that HAT inhibition by C646 increased the cytotoxic effect of gemcitabine on pancreatic cancer cell lines at 96 h. The researchers also confirmed that acetylation of H3K27 was inhibited and speculated that the development of gemcitabine resistance in pancreatic cancer was prevented, at least in part, by a HAT-dependent mechanism [83]. In addition, many other related studies have found that p300 inhibition enhances the sensitivity of drug-resistant tumour cells to chemotherapeutic agents by modulating HAT activity. For example, Mladek et al. found that CPI-1612 can effectively block RBBP4/p300 HAT activity by inhibiting the deposition of H3K27Ac in GBM cells, providing a new idea for the sensitisation of glioma to temozolomide (TMZ) [85]. In the study by Huang et al., p300 interference decreased the H3K27ac level, while a strain resistant to EZH2 inhibitors became sensitive. Subsequently, MLL1, which inhibits the binding of p300 to CBP and reduces the level of H3K27ac, was also assessed, providing research directions that could facilitate a more comprehensive understanding of p300 function and the development of tumour therapy [116].

In a study by Xu et al., p300 was found to promote acetylation of JMJD1A K421. The level of JMJD1A acetylation was increased in drug-resistant prostate cancer cell lines, and subsequent experiments also demonstrated that p300 inhibitors significantly inhibited the proliferation of resistant prostate cancer cell lines and increased the sensitivity of these tumour cells to AR antagonists [4]. Additionally, the chemotherapeutic drug 5-FU was found to reduce the ability of p300 to bind to chromatin; 5-FU induced H3 and H4 deacetylation and thus their degradation via lysosomes in a study by Du et al. The p300 expression profile was found to correlate with the resistance of colorectal cancer cells to 5-FU [86]. In response to DNA damage, p300 promotes DNA replication and repair through acetylation of H3 and/or H4 and induces chemotherapy and radiation therapy resistance in a variety of tumour types [117] (Table 2). Figure 5 shows a schematic of the modulation of resistance by p300.

### 4.5. p300 Regulates Tumour Metabolic Processes

p300 is a typical acetyltransferase and transcriptional coactivator that not only regulates transcription, apoptosis, and protein localisation, proliferation, migration, and invasion but also participates in metabolic processes. As we all know, tumour cell metabolism, means that tumour cells will acquire unique metabolic preferences depending on their tissue of origin, the degree of genetic changes, and the degree of interaction with hormones and systemic metabolites. That is, tumour occurrence is dependent on the reprogramming of cellular metabolism. p300 is involved in the regulation of tumour cell metabolism primarily as follows, for example, p300 can regulate glucose metabolism and fat metabolism. According to early studies on p300, it has a unique role in the control of metabolic processes as an acetyltransferase.

In studies related to prostate cancer, p300 was found to have a tumour-promoting effect and to regulate the expression of FASN, a regulator of lipid metabolism, by acetylating H3 in the FASN gene promoter. Therefore, p300 may be involved in the regulation of lipid metabolism [87]. Early studies of cellular lipid metabolism revealed that the stability of SREBP-1c is dynamically controlled by p300 acetylation and SIRT1 deacetylation, the latter being a mechanism specific to HepG2 cells [88]. A485, which regulates FOXO1 deacetylation and degradation via a proteasome-dependent pathway, has been shown to inhibit hepatic lipogenesis and glycoprotein production [118]. p300 and HDAC1 acetylate and deacetylate the catalytic subunit of adenosine monophosphate-activated protein kinase (AMPK); these proteins were found to enhance the interactions of the upstream kinase LKB1 when AMPK is deacetylated and lead to phosphorylation and activation of AMPK, resulting in lipolysis in human hepatocytes [107].

p300 can increase the H3K18Ac and H3K27Ac levels in the promoter regions of the metabolism-related genes PSAT1, ATIC and TALDO1 in Huh-7 cells, which in turn promotes glycolysis [71]. Similar studies have also demonstrated that B029-2, which regulates the acetylation of H3 K18 and K27, reduces the glycolytic capacity of hepatocellular carcinoma cells; thus, it has been explored as a potential p300-targeting drug for cancer treatment [71]. In addition to acetylation of nonhistone proteins, p300 catalyses 2-hydroxyisobutylation and lactylation of nonhistone proteins. Recent studies have also shown that p300 regulates glycolysis and lactate excretion by mediating the 2-hydroxyisobutyl conversion of ENO1 to induce acetylase activity, thereby regulating colon cancer metabolism [52]. Treatment with C646 also inhibits the metabolism of HepG2 (or Huh7) hepatocellular carcinoma cells via this mechanism [119]. p300 is also involved in the metabolic processes of immune cells. p300 helps macrophages take up extracellular lactate via monocarboxylic acid transporters (MCTs) and catalyses lactylation of HMGB1 in macrophages. C646 inhibits p300 acetylase activity and inhibits lactate-induced lactylation of HMGB1, two regulatory mechanisms by which p300 participates in macrophage metabolism [120]. In addition, deletion of the p300 gene to detect its effect on metabolic processes in serum induced changes in metabolites such as glutamate/glutamine and choline glycerate, and this finding was verified in HCT116 cells [121] (Table 2). Figure 6 shows a schematic diagram of the mechanisms of p300 in tumour metabolic regulation.

## 5. p300 Inhibitors

In many malignancies, p300 is considered a tumour promoter, a prospective target for tumour therapy, and a biomarker of tumour prognosis; as such, the development and application of p300 inhibitors is a priority.

Several p300-related inhibitors have been discovered over many years of scientific research and clinical exploration. For example, C646 not only controls the c-Met/Akt pathway and cyclin D1 expression but also induces the apoptosis of gastric cancer cells [2]. One study also found that C646 promotes the degradation of JMJD1A, which enhances the sensitivity of enzalutamide-resistant cells to C646, thus improving prostate cancer prognosis [4]. In addition, C646 enhances antitumour activity in colorectal cancer by blocking TRIB3 acetylation and promoting TRIB3 degradation [122]. C646 can also be used in the treatment of haematological tumours, and studies suggest that C646 may be a promising drug for widespread use in the treatment of AML [123,124]. Another inhibitor, CCS1477, inhibits prostate cancer cell proliferation and reduces the expression of AR and c-MYC regulatory genes, making it a promising novel prostate cancer treatment [78]. A high-throughput screen identified the histone acetylation compound L002 as a highly effective inhibitor of p300 in lymphoma cells and triple-negative breast cancer, making it a promising cancer treatment [107]. Both PU139 and PU141 are p300 inhibitors that significantly inhibit neuroblastoma growth and are effective against neuroblastoma [125]. Virtual ligand screening identified the p300 inhibitors anacardic acid [126], curcumin [127,128,129], chrysanthemum and isothiazole [130]. LTK-13 and LTK-14 are selective p300 inhibitors that inhibit the acetylation of p53 [131]. Other p300 inhibitors, such as RTK1 [132], LTK19 [131], and garcinol [131], have been discovered and employed in cancer therapy.

A485 is the most well-known selective p300 inhibitor, and its mechanism of action has been relatively clear for many years. It regulates tumourigenesis and progression by regulating transcription levels, thus affecting biological behaviour and immune function. p300 was also found to play a role in a variety of tumours and cell lines, such as prostate cancer, melanoma and various haematological tumours (multiple myeloma cell lines, AML cell lines and non-Hodgkin lymphoma cell lines) [70,133]. IIn other studies, four alkaloid p300 inhibitors were assessed, and their antitumour effects were validated in subsequent clinical studies [134]. In addition, some p300-related inhibitors, such as cyclopentenone prostaglandin, Y08197, CBP30, and B029-2, have been identified [71,135,136,137]. So far, two p300 inhibitors have been identified in clinical trials, including CCS1477 and FT-7051. Among them, CCS1477 has used in a phase I/II clinical trial in patients with metastatic castration prostate cancer. The other is FT-7051, which has started a phase one clinical trial in patients with metastatic castration resistant prostate cancer.

With the development of experiments related to p300 inhibitors, it is found that the curative effect of p300 inhibitors in the treatment of tumours is not a very ideal one. This is largely due to the fact that limited curative effect of p300 inhibitor monotherapy, thus, the majority of workers in scientific research try their best to explore a better treatment regime. Of these, combination therapy with inhibitors of p300 has an excellent therapeutic outlook. What is most notable is that some studies have begun to explore the therapeutic effect of the p300 inhibitor in combination with the PD-L1 antibody/anti-PD-L1 antibody. For example, in the treatment of colorectal cancer, the antitumour effect of C646 combined with PD-L1 antibody is obviously enhanced [122]. A485 combined with anti-PD-L1 antibody significantly inhibited androgen independent metastatic tumour growth in the setting of prostate cancer [70]. Based on these experiments, we can find that the p300 inhibitor in combination with the PD-L1 antibody/anti-PD-L1 antibody has a better therapeutic outlook in the treatment of various tumours. On the side, it can also be seen that the relationship between p300 and PDL1 and its related mechanism is worthy of investigation. Indeed, in some studies, there are also some explorations on its mechanism. For example, in Liu et al.’s study, p300 inhibitors can inhibit p300 from blocking the expression of PD-L1 induced by intrinsic and IFN-γ, and significantly improve the therapeutic effect of PD-L1 blocking on PCa.

p300 has been shown to play a key role in IFN-γ induced PD-L1 expression, and could be a good target for overcoming adaptive resistance induced by the PD-1/PD-L1 pathway [70]. In Gao et al.’ s research, it was confirmed that the acetylation and deacetylation of k263 site of PD-L1 were dynamically regulated by p300 and HDAC2, and modification of acetylation may promote PD-L1 translocation into the nucleus, and are involved in the regulation of IFN, NF-kB, MHC I, and other immune response gene expression by binding to DNA, thereby promoting tumours immune evasion [138]. there is not so much research on the relationship between p300 and PDL1 and its biological mechanism is relatively scarce, therefore, this will also be a research idea worthy of future study. We summarise some inhibitors of p300 and their main functions in Table 3.

## 6. Conclusions and Future Perspectives

The currently known functions of p300 are very broad; it participates in both cancer and nontumour diseases. As detailed in the above introduction to the roles of p300 in cancer, p300 plays a major role in posttranslational modification, which affects not only transcriptional regulation but also tumour cell proliferation, migration, invasion, and apoptosis, as well as the development of drug resistance and metabolic regulatory processes.

A large number of drugs targeting p300 have been studied in cell and animal experiments, but only two p300 inhibitors, such as CCS1477 and FT-7051, have been verified in clinical trials, which also reveals that it is extremely urgent for us to explore the clinical application value of more p300 inhibitors. In addition, in the experiment of p300 inhibitor, there are relatively few related studies to elaborate the treatment scheme of p300 inhibitor to maximise its efficacy, but novel explorations of p300 inhibitor treatment regimens remain. For example, the p300 inhibitor in combination with an anti-PD-L1 antibody is applied to experimental animals with tumours, greatly enhancing the anticancer efficacy of p300 inhibitor monotherapies.

Therefore, in future research, when discussing the mechanism and function of p300, researchers may focus on the following two aspects: on the one hand, at the level of posttranslational modification, we will continue to explore the relevant mechanism of catalytic action of p300 acetyltransferase; and on the other hand, we will focus on the existing posttranslational modification of p300 as a brand new target in order to develop more inhibitors of p300 and to further investigate a more scientific and efficient combination therapy scheme of p300 inhibitors and other anticancer drugs for clinical transformation.

## Figures and Tables

**Figure 1 biomolecules-13-00417-f001:**
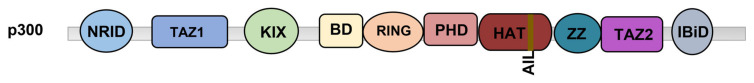
Schematic of the basic structure of p300, which contains the N-terminal nuclear receptor interaction domain (NRID), transcriptional junction zinc finger 1 (TAZ1), kinase-induced CREB interaction domains (KIX), bromine domain (BD), RING domain, plant homology domain (PHD), HAT domain, Z-type zinc finger domain (ZZ), TAZ2 domain and IBiD domain.

**Figure 2 biomolecules-13-00417-f002:**
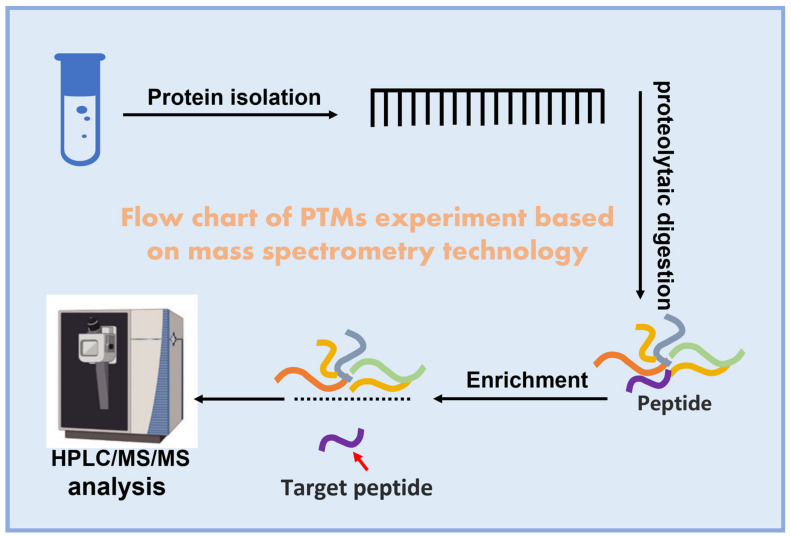
Diagram of experiments to detect posttranslational modifications: preparation of purified protein samples, digestion of proteins into peptides, enrichment of specific PTM peptides using appropriate methods, and HPLC/MS/MS analysis of specific enriched PTM peptides.

**Figure 3 biomolecules-13-00417-f003:**
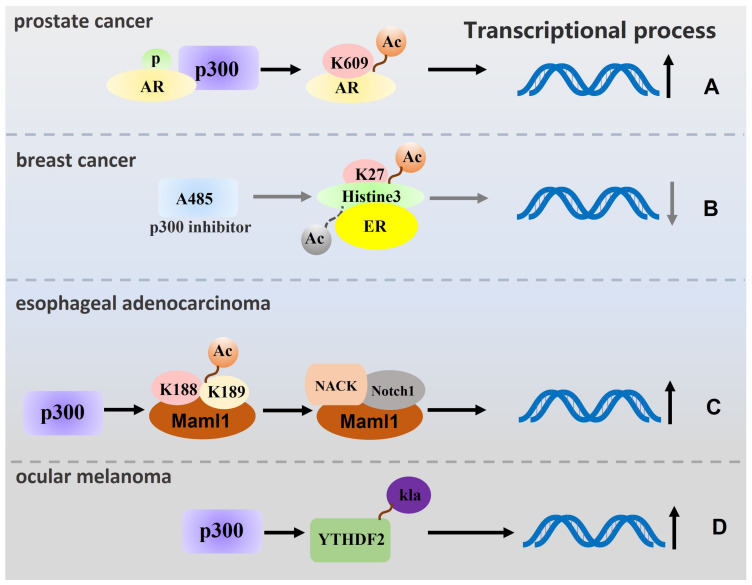
Schematic representation of transcription-related pathways regulated by p300. (**A**) The p300 gene recognises phosphorylated AR in prostate cancer and promotes AR acetylation in K609, thereby promoting transcription. (**B**) A485, an inhibitor of p300, can reduce the level of transcription of highly expressed genes by reducing the level of H3K27ac in particular genes (such as ER). (**C**) In esophageal adenocarcinoma, p300 acetylates K188 and K189 of Maml1, and recruits NACK to the ternary complex of Notch1, resulting in the initiation of transcription. (**D**) p300 can catalyse the lactate of YTHDF2 promoter in ocular melanoma cells and then regulate the transcription process. (Remarks: The grey arrow is the related process in which p300 inhibitors participate in regulation, and the dotted line is the process in which modification types are removed).

**Figure 4 biomolecules-13-00417-f004:**
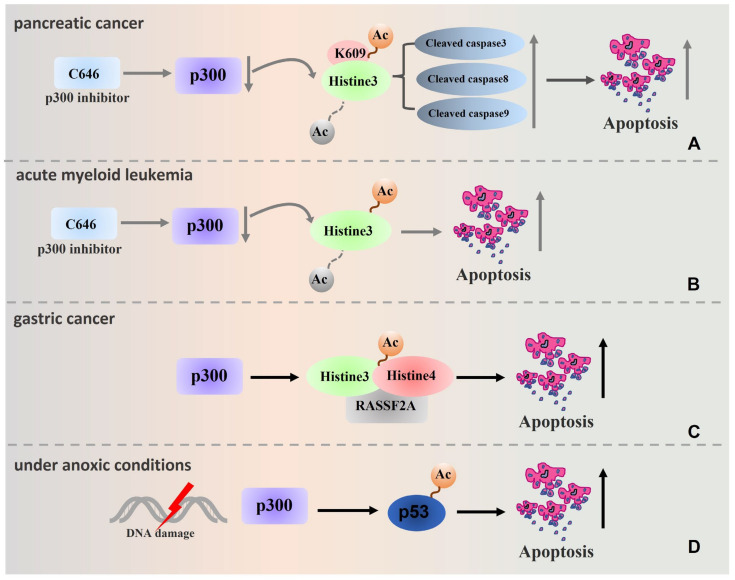
Schematic representation of pathways involved in the regulation of apoptosis by p300. (**A**) In pancreatic cancer studies, C646 treatment was followed by interference with p300 followed by inhibition of acetylation of H3K27, leading to increased expression of apoptosis-related proteins such as cleaved caspase 3, 8 and 9 and promoting cell apoptosis. (**B**) C646 treatment leads to apoptosis in AML1-ETO positive AML cell line and primary parental cells, which may be associated with p300-mediated histone acetylation. (**C**) p300 is involved in the acetylation of the Histones H3 and H4 on RASSF2A promoter and regulates the expression of RASSF2A, thus inducing apoptosis of gastric cancer cells. (**D**) p300 acetylates p53 and regulates apoptosis in the presence of hypoxia or DNA damage (Remarks: The grey arrow is the related process in which p300 inhibitors participate in regulation, and the dotted line is the process in which modification types are removed).

**Figure 5 biomolecules-13-00417-f005:**
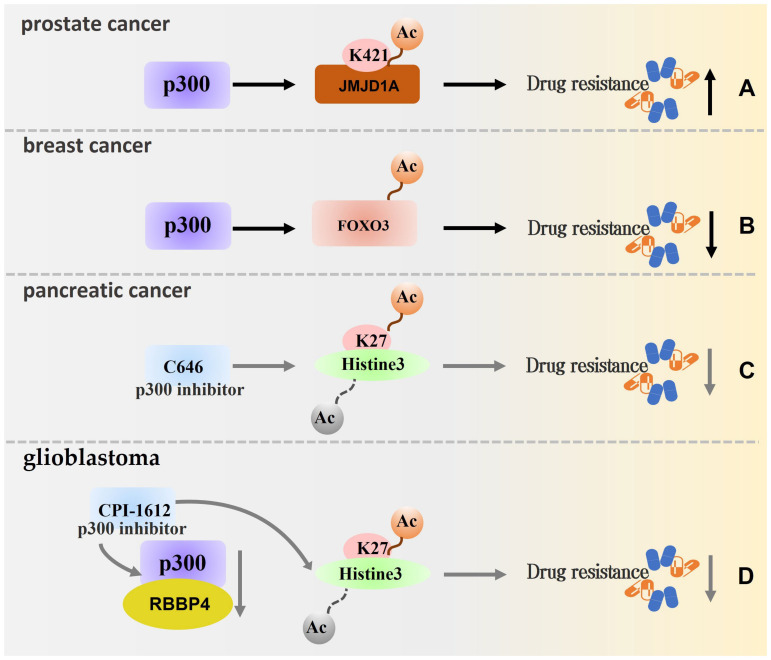
Schematic of the modulation of resistance by p300. (**A**) p300 promoted the acetylation of JMJD1A k421, and the acetylation level of JMJD1A increased in drug-resistant prostate cancer cell lines. (**B**) p300 mediates the acetylation of FOXO3 and enhances the sensitivity to lapatinib in the treatment of breast cancer. (**C**) C646 inhibits the acetylation of H3K27 to a degree that inhibits gemcitabine resistance in the treatment of pancreatic cancer. (**D**) After treatment with p300 inhibitor CPI-1612, the activity of RBBP4/p300 HAT was effectively blocked, which made the treatment of glioblastoma sensitised to temozolomide (TMZ). (Remarks: The grey arrow is the related process in which p300 inhibitors participate in regulation, and the dotted line is the process in which modification types are removed).

**Figure 6 biomolecules-13-00417-f006:**
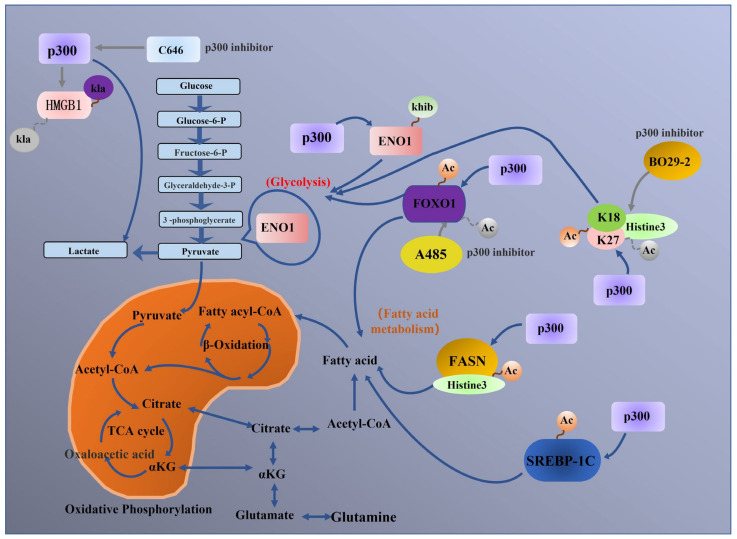
Schematic diagram of the mechanisms of p300 in tumour metabolism regulation. p300 catalyses many different types of posttranslational modifications, such as acetylation, β-hydroxyisobutyrylation and lactylation, in tumour cells to modulate glycometabolism and lipid metabolism. (Remarks: The grey arrow is the related process in which p300 inhibitors participate in regulation, and the dotted line is the process in which modification types are removed).

**Table 1 biomolecules-13-00417-t001:** Summary of common posttranslational modification types, modification sites, regulatory factors, enrichment modes, and mass shifts in mass spectrometry.

PTM	Modification Sites	Regulatory Factors	Enrichment Modes	Mass Shif (Da)
phosphorylation	serine, threonine, tyrosine	protein kinase; protein phosphatase	titanium dioxide	80
methylation	lysine, arginine	PKMTs, PRMTs	lysine methylation antibody, arginine methylation antibody	14, 28, 42
oxidative modification	methionine, tryptophan	ROS	oxPTMs antibody	4, 16, 32
ubiquitination	lysine	Ubiquitin activator E1, ubiquitin binding enzyme E2 and ubiquitin ligase E3	lysine ubiquitination antibody	>1000, 114 after trypsinisation
acetylation	lysine	p300; SIRT1, SIRT3, SIRT6	lysine acetylated antibody	42
propionylation	lysine	p300, NATs (GCN5, PCAF), MYSTs (MOF, MOZ, HBO1); SIRT1-3	lysine propionylation antibody	56
butyrylation	lysine	p300, HBO1; SIRT1-3	lysine butyrylation antibody	70
malonylation	lysine	SIRT5	lysine malonylation antibody	86
glutarylation	lysine	p300, GNATs (GCN5);	lysine glutarylation antibody	114
		SIRT5, SIRT7		
hydroxybutyrylation	lysine	p300	lysine hydroxybutyrylation antibody	86
2-hydroxyisobutyrylation	lysine	p300, MYSTs (ESa1p, Tip60);	lysine 2-hydroxyisobutyrylation antibody	86
		HDAC1-3, Rpd3p, CobB		
succinylation	lysine	GNATs (GCN5, HAT1), CPT1A, KGDHC; SIRT5, SIRT7	lysine succinylation antibody	100
crotonylation	lysine	p300; HDAC-3, SIRT1-3	lysine crotonylation antibody	68
lactylation	lysine	p300; HDAC1-3, SIRT1-3	lysine lactylation antibody	72

**Table 2 biomolecules-13-00417-t002:** Summary of the role of p300-mediated posttranslational modification in tumours biological function.

Biological Function	Tumour Type	Name of Protein	Modification Type	Effects on Tumours	Regulatory Factors	References
transcriptional control	prostate cancer	AR	Acetylation	Promoting AR acetylation at AR K609 site and promoting transcription process	p300	[62]
transcriptional control	breast cancer	Histone3	Acetylation	Enhance the level of H3K27ac in ER to enhance the transcription level of highly expressed genes	p300	[64]
transcriptional control	esophageal adenocarcinoma	Maml1	Acetylation	Promote the acetylation of K188 and K189 of Maml1 and promote the transcription process	p300	[66]
transcriptional control	hepatocellular carcinoma	Histone3	Acetylation	p300 inhibitor B029-2 can block the binding of H3K18ac and H3K27Ac with related promoters and hinder the transcription process	p300	[71]
transcriptional control	ocular melanoma	Promoter of YTHDF2	Lactylation	p300 catalyses histone lactate and promotes its transcription	p300	[73]
proliferation, migration and invasion	colorectal cancer	MTA2	Acetylation	Promotes colorectal cancer cell migration and invasion	p300	[74]
proliferation, migration and invasion	liver cancer	Histone3, PGK1	Acetylation	Promotes proliferation and enhances metastatic capacity of HCC cells	p300	[71,75]
proliferation, migration and invasion	nasopharyngeal carcinoma	Smad2 and Smad3	Acetylation	Induces EMT and promotes CNE-2 invasion and migration in human nasopharyngeal carcinoma cells	p300	[76]
proliferation, migration and invasion	prostate cancer	AR	Acetylation	Stabilises AR and promotes the proliferation of prostate cancer cells	p300	[77,78]
proliferation, migration and invasion	cervical cancer	HNRNPA1	Crotonylation	Promotes HeLa cell proliferation, migration, and invasion	p300	[79]
proliferation, migration and invasion	breast cancer	MRTF-A	Acetylation	Increases the viability and promotes the migration of breast cancer cells	p300	[67]
proliferation, migration and invasion	osteosarcoma	JHDM1A	Acetylation	Acetylation of JHDMIA can inhibit the proliferation and invasion of osteosarcoma HOS cells	p300	[11]
proliferation, migration and invasion	T-cell acute lymphoblastic leukaemia	Notch3	Acetylation	Promotes proliferation of AML cell lines	p300, HDAC1	[80,81,82]
cell apoptosis	pancreatic cancer	Histone	Acetylation	After being treated with C646, it can inhibit the acetylation of H3K27, and then regulate the apoptosis of pancreatic cancer cells.	p300	[83]
cell apoptosis	gastric cancer	Promoter of RASSF2A	Acetylation	Participate in the acetylation of RASSF2A promoter, and then induce apoptosis of gastric cancer cells	p300	[59]
drug resistance formation	breast cancer	FOXO3	Acetylation	Lapatinib is a chemotherapy drug for breast cancer. p300 mediates the acetylation of FOXO3 and enhances its sensitivity to Lapa-tinib	p300	[84]
drug resistance formation	glioblastoma	Histone3	Acetylation	CPI-1612 can inhibit the deposition of H3K27Ac in GBM cells, which provides a basis for the sensitisation of TMZ	p300	[85]
drug resistance formation	prostate Cancer	JMJD1A	Acetylation	The acetylation level of JMJD1A increased in drug-resistant prostate cancer cell lines	p300	[4]
drug resistance formation	colorectal cancer	Histone3/4	Acetylation	p300 expression profile was found to be related to the resistance of colorectal cancer cells to 5-FU	p300	[86]
cellular metabolism	prostate Cancer	Promoter of FASN	Acetylation	Acetylating H3 in FASN gene promoter to regulate lipid metabolism	p300	[87]
cellular metabolism	liver cancer	SREBP-1c	Acetylation	The stability of SREBP-1c is dynamically controlled by p300 acetylation and SIRT1 deacetylation, which is an important mechanism of lipid regulation in hepatocellular carcinoma cells	p300, SIRT1	[88]
cellular metabolism	carcinoma of colon	ENO1	2- hydroxyisobutyration	Catalysed the acylation of ENO1 2- hydroxyisobutyrate, and then enhanced its glycolytic ability and lactic acid excretion	p300	[52]

**Table 3 biomolecules-13-00417-t003:** Summary of representative p300 inhibitors and their main functions.

Name	Structure	Main Function	Clinical Trial Landscape	References
C646	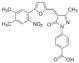	C646 regulates tumour cell drug resistance in cancers such as prostate cancer. C646 can enhance the antitumour activity in colorectal cancer by blocking the acetylation of TRIB3 and promoting the degradation of TRIB3. C646 can block the tumour cell cycle, induce tumour cell apoptosis, and inhibit tumour cell proliferation and growth in lung cancer and melanoma	N/A	[108,122,130]
A485	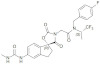	A485 regulates antitumour activity in a variety of solid tumours such as melanoma and prostate cancer and in haematological tumours (such as multiple myeloma, acute myeloid leukaemia, and non-Hodgkin lymphoma). Combination of an anti-PD-L1 antibody with A485 can further enhance its antitumour efficacy	N/A	[70,124,133,139]
CCS1477	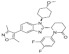	CCS1477 can exert anticancer effects in prostate cancer by decreasing AR expression and regulating C-MYC gene expression	Phase I/IIa clinical trial for patients with advanced solid tumours	[78]
B029-2	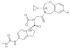	B029-2 can inhibit glycolysis, induce tumour cell cycle arrest, and inhibit tumour cell proliferation, thus delaying the malignant progression of liver cancer	N/A	[71]
B026	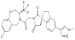	B026 can significantly inhibit the expression of MYC in leukaemia cell lines and lymphoma cell lines and significantly decrease the level of H3K27Ac, and it has better effects and stability than A485. In addition, B026 can inhibit the proliferation of prostate cancer Cells	N/A	[140]
CCT077791	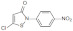	CCT077791 treatment for 24 h inhibited colon cancer cell proliferation by reducing the H3 and H4 acetylation levels in HCT116 and HT29 cells	N/A	[105]
CBP30	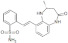	CBP30 can impair Treg differentiation and inhibit their function, which in turn enhances antitumor immune function and exerts antitumour effects. CBP30 also inhibits human Th17 responses	N/A	[137,141]

## Data Availability

Not applicable.

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
