# Peer review of "Effects of the Acetyltransferase p300 on Tumour Regulation from the Novel Perspective of Posttranslational Protein Modification"

_biomolecules, 2023, doi:10.3390/biom13030417_

Round 1
Reviewer 1 Report
The authors in the review “Effects of the acetyltransferase p300 on tumour regulation from the novel perspective of protein posttranslational modification“ are reporting all the revelant discoveries about the function of p300.
The manuscript covers wide range of publications as there is a lot research done with p300. It covers all the relevant data for the p300 function in the tumors.
However, in some part it is quite hard to read it and follow all the information mention in the text. It should be added some text with authors prospective and summaries of the mention research to make it a bit easier to follow. Tables are good to summaries all the data from references so there is no need to cover it all over again in the text.
Minor remarks:
Figure 1. – all the domains should be named, all the abbreviation explained in the description of the figure.
Figure 2. – 2A should be bigger and there is no need for 2B as there is whole table 1 mentioning same data. In table 1 should be added column remarking which of the PTM refers to p300.
Figure 3. – Illustrations are not in the good proportions, it is hard to distinguish proteins from PMTs. Name of the tumors should be added in the figure. It is only mention in the description of the figure so it’s hard to read from the figure. My suggestion is to add the table for 4.1 – 4.5
Figure 5. – It is not clear the difference in the figure description between A and B.
It is not clear if there is any clinical trial with p300 inhibitors. It should be stated in the table 3 if there is any.
As there are many references in the manuscript, authors should go thourally through them again before resubmission.
Author Response
Dear reviewer,
First of all, thank you very much for your precious time to read my article and put forward valuable comments and opinions. I will try my best to reply to your suggestions one by one, hoping to get your approval.
Point 1: In some part it is quite hard to read it and follow all the information mention in the text. It should be added some text with authors prospective and summaries of the mention research to make it a bit easier to follow. Tables are good to summaries all the data from references so there is no need to cover it all over again in the text.
Response 1: I can't agree with you more. At your suggestion, I will also add some contents covering my own prospective of p300 and summaries of the full text, so as to make the full text easier to read. There is some overlap between the table and the text. I will try my best to correct and adjust it.
Point 2: Figure 1. – all the domains should be named, all the abbreviation explained in the description of the figure.
Response 2: In the Figure 1, I neglected some details in the description of p300 domain, which will be well supplemented.
Point 3: Figure 2. – 2A should be bigger and there is no need for 2B as there is whole table 1 mentioning same data. In table 1 should be added column remarking which of the PTM refers to p300.
Response 3: Following your suggestion, I will resize Figure 2A to be larger and delete Figure 2B. And I will show the type of p300 catalytic post-translation modification mentioned in Figure 2B in table 1.
Point 4: Figure 3. – Illustrations are not in the good proportions, it is hard to distinguish proteins from PMTs. Name of the tumors should be added in the figure. It is only mention in the description of the figure so it’s hard to read from the figure. My suggestion is to add the table for 4.1 – 4.5
Response 4: I'll readjust the scale of Figure 3, and I'll revise the performance of PTM and protein to distinguish them better. And I will also add the tumor type in Figure 3. I will try my best to add the table of 4.1-4.5.
Point 5: Figure 5. – It is not clear the difference in the figure description between A and B.
Response 5: The description of Figure 5 is not clearly expressed, so I will correct it again accurately.
Point 6: It is not clear if there is any clinical trial with p300 inhibitors. It should be stated in the table 3 if there is any.
Response 6: I will add relevant information about whether p300 inhibitors have been tested in table 3, so that readers can have a more comprehensive understanding of p300 inhibitors.
Point 7: As there are many references in the manuscript, authors should go thourally through them again before resubmission.
Response 7: I will check the accuracy and authenticity of the references again.
Finally, thank you again for your valuable advice. I will try my best to correct the content according to your suggestion.
kind regards,
Qingmei Zeng
Reviewer 2 Report
This reviewer suggests that the manuscript is edited by someone with full professional proficiency in English before resubmitting.
The manuscript needs significant improvement in writing. This reviewer had to guess the information in the manuscript the authors tried to convey and stopped the reading after reaching the Table 1 and Figure 1.
This reviewer is very interested in the topic of the manuscript. From the long list of the references, this reviewer assumed the manuscript covers a lot of information. However, it is too hard to catch the meaning of the authors within a limited time.
Author Response
Dear reviewer,
First of all, thank you very much for your precious time to read my article and put forward valuable comments and opinions. I will try my best to reply to your suggestions one by one, hoping to get your approval.
Point 1: This reviewer suggests that the manuscript is edited by someone with full professional proficiency in English before resubmitting.
Response 1: I can't agree with you more. Following your suggestion, before we resubmit the manuscript, I will give it to someone with complete professional English level for perfect language editing.
Point 2: The manuscript needs significant improvement in writing. This reviewer had to guess the information in the manuscript the authors tried to convey and stopped the reading after reaching the Table 1 and Figure 1.
Response 2: Thank you for your valuable comments. I will improve the writing of the full text again. I am really sorry for your reading difficulties due to my poor writing ability and lack of language problems. However, I will really try my best to cultivate my writing ability and strive to contribute my own language editing ability in the following days. Of course, I will also follow your suggestion and find English experts to help me with my language notes. I hope you can give me the opportunity to seriously correct my writing and language editing.
Finally, thank you again for your valuable advice. I will try my best to correct the content according to your suggestion.
kind regards,
Qingmei Zeng
Reviewer 3 Report
Zeng et al., explained the activity of the co-transcription factor p300 in tumour development. They explained the structure of the molecule, modifications, interacting proteins and its involvement in the tumour metabolism regulation. The authors also introduced the chemicals involved in the activity of p300 inhibition.
They cited an adequate number of references and the structure of the manuscript is good for publishing in the indicated journal.
I suggest the authors, if they can add some comments on a cutting-edge therapeutic method, such as PD-L1, on their review.
Author Response
Dear reviewer,
First of all, thank you very much for your precious time to read my article and put forward valuable comments and opinions. I will try my best to reply to your suggestions one by one, hoping to get your approval.
Point 1: I suggest the authors, if they can add some comments on a cutting-edge therapeutic method, such as PD-L1, on their review.
Response 1: I totally agree with your suggestion. I will try to add some comments on cutting-edge treatment methods, such as PD-L1, to my review by consulting relevant knowledge.
Finally, thank you again for your valuable advice. I will try my best to correct the content according to your suggestion.
kind regards,
Qingmei Zeng
Round 2
Reviewer 2 Report
It is not the grammar problems of sentences, but the ways of writing that might lead to confusion, especially for those new to the field.
It is better to give some brief explanations in the captions so that the reader can understand the figures easily without referring to other resources. For example, there are a lot of arrows in the figures, what are the differences between solid and dotted lines, dark and gray…?
Author Response
Dear reviewer,
First of all, thank you very much for your precious time to read my article and put forward valuable comments and opinions. I will try my best to reply to your suggestions one by one, hoping to get your approval.
Point 1: It is better to give some brief explanations in the captions so that the reader can understand the figures easily without referring to other resources. For example, there are a lot of arrows in the figures, what are the differences between solid and dotted lines, dark and gray…?
Response 1: We thank you for your feedback on the writing, and we apologize for the poor writing. At the same time, thank you again for your valuable comments on my manuscript. To this end, I revised the whole manuscript text accordingly. I will try my best to correct my writing and follow your advice. I will add explanatory paragraphs where appropriate, and add missing details in the picture description, so that readers can understand the contents of the full text more easily. As I seldom come into contact with this field and lack more valuable experience, I really gained a lot under your advice. I will always insist on working hard on my writing style. Therefore, I hope you can give me the opportunity to further improve and learn in this field.
Finally, thank you again for your valuable advice. I will try my best to correct the content according to your suggestion.
kind regards,
Qingmei Zeng